# Genetics Matters: Voyaging from the Past into the Future of Humanity and Sustainability

**DOI:** 10.3390/ijms23073976

**Published:** 2022-04-02

**Authors:** Acga Cheng, Jennifer Ann Harikrishna, Charles S. Redwood, Lei Cheng Lit, Swapan K. Nath, Kek Heng Chua

**Affiliations:** 1Institute of Biological Science, Faculty of Science, Universiti Malaya, Kuala Lumpur 50603, Malaysia; acgacheng@um.edu.my (A.C.); jennihari@um.edu.my (J.A.H.); 2Centre for Research in Biotechnology for Agriculture, University of Malaya, Kuala Lumpur 50603, Malaysia; 3Radcliffe Department of Medicine, University of Oxford, John Radcliffe Hospital, Oxford OX3 9DU, UK; charles.redwood@cardiov.ox.ac.uk; 4Department of Physiology, Faculty of Medicine, Universiti Malaya, Kuala Lumpur 50603, Malaysia; lleicheng@um.edu.my; 5Oklahoma Medical Research Foundation, Oklahoma City, OK 73104, USA; 6Department of Biomedical Science, Faculty of Medicine, Universiti Malaya, Kuala Lumpur 50603, Malaysia

**Keywords:** agriculture, biodiversity, heredity, gene-editing, genetic technologies, medicine, sustainability

## Abstract

The understanding of how genetic information may be inherited through generations was established by Gregor Mendel in the 1860s when he developed the fundamental principles of inheritance. The science of genetics, however, began to flourish only during the mid-1940s when DNA was identified as the carrier of genetic information. The world has since then witnessed rapid development of genetic technologies, with the latest being genome-editing tools, which have revolutionized fields from medicine to agriculture. This review walks through the historical timeline of genetics research and deliberates how this discipline might furnish a sustainable future for humanity.

## 1. A Trip down Memory Lane: One and a Half Centuries into the Intriguing Study of Heredity

Gregor Mendel, recognized as the Father of Modern Genetics, was an Austrian monk who established the foundational principles of heredity through his breeding experiments on the common pea (*Pisum sativum*) and coined the terms dominant and recessive [1]. Mendel deemed that peas were a suitable model system due mainly to their distinct, constant differentiating characteristics and their hybrids yielding perfectly fertile progeny [2,3]. After eight years of tedious pursuit, he finally published his work entitled “Experiments on Plant Hybridization” in 1866, proposing the principles of uniformity, segregation, and independent assortment [1]. Compared to his contemporary Charles Darwin who developed the Theory of Evolution, Mendel’s work was not widely known until the 1900s, and its relevance fell in and out of favour as genetic theory continued to develop [2,4]. In 1882, chromosomes were first described by Walter Flemming, the founder of the science of cytogenetics who pioneered the study of mitosis [5]. The chromosomal theory of heredity, however, was only established in 1910 when Thomas Morgan discovered sex chromosome inheritance through his breeding analysis on millions of wild-type red-eyed and white-eyed fruit flies (*Drosophila melanogaster*) [6]. Morgan’s findings confirmed Mendel’s principles of heredity, and that genes are located on chromosomes [7].

A gene, the basic unit of heredity, is made up of deoxyribonucleic acid (DNA) which was first characterized by Friedrich Miescher about 150 years ago in 1871 [8]. While examining proteins in leucocytes, Miescher obtained a novel substance in the nuclei that differed fundamentally from proteins, which he termed nuclein [9]. Like Mendel, while Miescher’s discovery was well ahead of its time, unfortunately nuclein remained mostly unknown until the interest in the DNA molecule was revived around the mid-1940s. This was when Oswald Avery and his colleagues published the first evidence of DNA, instead of protein, as the carrier of genetic information in their transformation experiments using pneumococcus bacteria [10]. This work also led to the revelation that the three-dimensional structure of DNA exists as a double helix, deciphered by James Watson and Francis Crick in 1953. Nucleotide, the basic building block of DNA, is composed of a five-carbon sugar molecule (i.e., deoxyribose), a phosphate group, and one of the four nitrogen bases, specifically adenine (A), cytosine (C), guanine (G), and thymine (T) which provide the underlying genetic basis (i.e., the genotype) for informing a cell what to do and what kind of specialized cell to become (i.e., the phenotype). The Watson–Crick model has been both highly acclaimed and controversial, where the latter stems largely from the fact that their work was directly dependent on the research of several scientists before them, including Maurice Wilkins and Rosalind Franklin [11,12].

In the early 1960s, the genetic code, which consists of 64 triplets of nucleotides was decoded by Nirenberg, et al. [13], followed by the establishment of the central dogma of biology which explains the flow of genetic information from gene sequence to protein product through three fundamental processes, namely replication, transcription, and translation. Nevertheless, the first violation of central dogma was reported in less than a decade later in 1970 when David Baltimore and Howard Temin discovered reverse transcriptase in retroviruses, demonstrating the possibility of the reverse transmission of genetic information from ribonucleic acid (RNA) to DNA. Their discovery has revolutionized molecular biology and formed the cornerstone of cancer biology and retrovirology [14].

Initial efforts to sequence a gene were rather cumbersome and time consuming, for example, it took months to sequence a mere 24-base pair lactose operon of *Escherichia coli* using the Maxam–Gilbert sequencing that involved extensive use of hazardous chemicals [15]. Invented by Allan Maxam and Walter Gilbert in the mid-1970s, the Maxam–Gilbert sequencing method involves chemical alteration of DNA and subsequent cleavage at specific bases, which necessitates radioactive labelling at one end and purification of the DNA fragment of interest [15,16]. The dawn of rapid sequencing began in 1977 when the chain termination method, better known as Sanger sequencing, was developed based on the process of DNA replication [17]. Pioneered by Frederick Sanger, the technique was used to sequence the first DNA genome, the bacteriophage *ϕ*X174, often used as a positive control genome in sequencing labs around the world since its completion [17,18]. Another notable invention from that time period is polymerase chain reaction (PCR), a technique for generating millions of copies of a specific section of DNA [19]. Invented by Kary Mullis in 1983, the technique has since been employed for a variety of applications, including decoding the human genome, preserving animals and coral reefs, and, most recently, detecting COVID-19 [20].

The early 21st century saw the rise of the next-generation DNA sequencing (NGS), when booming sequencing companies were hosting their personalized technologies, with the initial “big three” platforms being Roche/454, Life Technologies/SOLiD, and Illumina/Solexa. In contrast to the Sanger method, which only allows for the sequencing of a single DNA fragment at a time, NGS can sequence millions of fragments in a single run [21]. It is worth noting that the sequencing cost has dropped dramatically over the years since the invention of automated sequencing protocols. Take the human genome, for example. Sequencing costs have decreased from approximately USD 3 billion for the first 3.2 gigabyte genome in 2002 to as low as about USD 1000 for a genome today. The first next-generation sequencer, the 454 system, was introduced in 2005 by Jonathan Rothberg and his colleagues who demonstrated how the genome of a parasitic bacterium *Mycoplasma genitalium* was sequenced in a single run utilizing the emulsion PCR technique. Likewise, the team sequenced the genome of James Watson, piloting the prevalent personalized genomics [22]. Since then, several sequencers have been developed, including the Solexa 1G (or Genome Analyzer 1), HiSeq 2000, and ultra-high-throughput systems like the HiSeq 4000 and NovaSeq 6000 [22,23].

During the 2010s, more far-reaching sequencing technologies have been developed, from semiconductor chips to nanoballs, all of which provide variable impacts upon what studies are feasible and on the market at large [18,24]. The advent of gene-editing technologies, such as clustered regularly interspaced short palindromic repeats (CRISPR), has transformed many fields in the 21st century, particularly medicine. One recent example is the development of the rapid and accurate CRISPR-Cas12-based detection of the betacoronavirus severe acute respiratory syndrome (SARS)-CoV-2, which has caused over one million deaths worldwide since the outbreak began in December 2019 [19,25]. Gene editing is not a new phenomenon; techniques for editing and knocking out genes have been available since the 1980s, when gene-editing technology was initially developed and introduced [26]. The use of genome editing has expanded the potential of therapeutic technologies, with induced pluripotent stem cells (iPSCs) being a recent example, which generate new models and treatments for a variety of disorders, including Alzheimer’s and Parkinson’s diseases [27].

The study of heredity for the past one and a half centuries has been fascinating, with numerous notable successes summarised in Figure 1. Within 100 years, scientists in the modern era figured out how DNA works, and designed machines that could read it, and, more recently, tools that could edit it. Nevertheless, some of these successes are bound to be controversial, especially the gene-editing technology. The study of heredity has come a long way since Mendel’s work, but the quest for a heathier and more sustainable future through modern genetics continues.

## 2. How Has the Cracking of Genetic Code Improved Life on Earth?

Life has existed on Earth for approximately four billion years, albeit the genetic code was only decoded in the 1960s after 99% of human history has been documented [28]. Within the last half-century the world has witnessed tremendous discoveries in all critical areas of life sciences, from medicine to agriculture. The human genome was completed in the early 2000s, around the same time as a handful of model organisms, including arabidopsis (*Arabidopsis thaliana*), rice (*Oryza sativa*), and fruit fly (*Drosophila melanogaster*) (Figure 1). After the birth of next-generation sequencing (NGS) technologies in the mid-2000s, hundreds of other organisms had their genomes completely sequenced, and millions of genes have been annotated, be they genes responsible for severe diseases in humans or genes conferring resistance and tolerance in crops [29]. This is especially true for microorganisms with small genome sequences, such as viruses and single-celled organisms (bacteria and protozoa), where the number of genomes being sequenced for these organisms has been exploding [30]. Both partial and complete COVID-19 genome sequences were obtained in the first two months of the epidemic [31]. However, like all other viruses, the SARS-CoV-2 undergoes mutation or small changes in its genome, demonstrating that the virus is evolving [32]. To allow the human genetics community to share important outcomes of the genetic determinants of COVID-19 susceptibility and severity, the COVID-19 Host Genetics Initiative [33] was established in spring 2020 and the genetic association results of several gene clusters (such as TYK2, DPP9, and the OAS1/2/3) were publicly released in January 2021 [34].

Although the sequencing for multicellular organisms has slightly lagged behind, a handful of massive sequencing projects are actively ongoing, including the Earth BioGenome Project that aims to sequence the genomes of all 1.5 million known eukaryotic species, and also the Darwin Tree of Life Project, which seeks to obtain the code of 66,000 species of sequence from every animal, plant, and fungus in the United Kingdom over the course of a decade [35,36]. To date, there are more than 16,000 sequenced genomes of eukaryotes available in the public domain (https://www.ncbi.nlm.nih.gov). Table 1 presents some examples of well-annotated genomes of multicellular organisms evolved in the primary eukaryotic kingdoms since the 2000s.

An organism’s genetic code is made up of merely four bases—A, C, G, and T, but just a change in a single base, frequently known as single nucleotide polymorphism, among thousands of bases can potentially lead to changes in protein structures and functions, impacting one or various traits of an organism. The abnormal changes in the DNA of a gene are termed gene mutations, which may have little to no noticeable effects, or can considerably affect cells in numerous ways [86]. Some mutations cause a gene to be turned on, making more of the protein than usual, while a small percentage of mutations has been found to cause genetic disorders [86,87]. For instance, a mutated version of the beta-globin gene that helps make haemoglobin causes sickle cell anaemia [88]. Recently, a robust sequence-resolved benchmark set for detection of both false positive and false negative germline large insertions and deletions has been developed [89]. The modern genetic modification, interchangeably known as genetic engineering, is the process of altering the genetic makeup of an organism using recombinant DNA (rDNA) technology. A gene (sometimes two or more) from a species is isolated, spliced into a vector with the aid of restriction enzymes, and then introduced into the host species, creating a “transgenic” organism, called a genetically modified organism (GMO), with desirable characteristics [1]. The first genetically engineered animal (i.e., mouse; *M. musculus*) and genetically engineered plant (i.e., tobacco; *Nicotiana tabacum*) were produced in 1974 and 1983, respectively. The inception of genetically engineered technology initially sparked concern from various parties, including governments, scientists, and the media, over its potential adverse effects in human health or ecosystems worldwide [90]. With the establishment of a safe and practical guide to rDNA research in 1975, the technology has continued to advance rapidly, impacting medicine, agriculture, and biodiversity [91,92].

### 2.1. Medicine

Since the completion of the Human Genome Project, multitudinous pieces of research have been underway into human genetic diseases, with the most common one being cancer. The International Cancer Genome Consortium (ICGC) was launched in 2008 to generate genetic data for about 50 most common cancer types and/or subtypes across the globe (https://icgc.org). In 2006, treatments targeting specific molecular abnormalities were made available for certain types of cancer, such as melanoma [93] and lung cancer [94], making some of these chronic illnesses manageable and possibly curable. This was made possible when Druker et al. [95] developed imatinib (or Gleevec), a drug with high efficiency in treating chronic myelogenous leukaemia by targeting the unique molecular abnormality. To date, more than 1000 human genetic tests are practicable, and some enable embryos created from in vitro fertilization to be screened for the genetic mutations that cause genetic disorders such as sickle cell disease and cystic fibrosis [96,97].

Gene therapy, which targets faulty or missing genes to treat disease, is at the forefront of modern medicine [98]. While gene therapy is currently being tested only for terminal diseases like haemophilia and AIDS, this innovation has shown promising progress during the past two decades, with a handful of notable successes including treatments for the X-linked severe combined immunodeficiency and the inherited blindness Leber’s congenital amaurosis 2, which are caused by mutations in the interleukin-2 receptor γ chain (*IL2RG*) and the retinal pigment epithelium-specific 65 kDa protein (RPE65) genes, respectively [99,100]. Gene therapy trials, however, can raise the risk of severe side effects which can lead to death [101,102], and this still-evolving molecular medicine may require many more years of testing to be proven effective and safe for most conditions [98,103].

Different types of gene-targeting vectors have been designed to elucidate gene function in vivo, from point mutations and insertions to gene deletions [104]. Before individual genome sequencing becomes routine, DNA (or gene) chips can be considered as one of the critical pharmacogenetics technologies. Featuring a tiny DNA microarray, gene chips reveal the level of activation of particular genes and assess a patient’s genetic suitability for certain drugs [105]. Technological advancements in sequencing have facilitated the integration of pharmacogenetics (or pharmacogenomics) in clinical diagnostics, allowing doctors to prescribe medication based mainly on their individual patient’s genetics rather than factors like age and body mass [106]. With the information on how genes influence the response of different individuals on the same drug or medication, treatments can be selected more accurately, and the significant side effects of a specific drug on certain individuals can be avoided [107,108]. Precision approaches are promising in protecting population health and addressing global health landscape challenges (such as the spread of SARS-CoV-2 infections), but they should be complementing rather than replacing the efforts to strengthen public health infrastructure [108]. Recently, several genetic markers associated with SARS-CoV-2 and COVID-19 disease severity have been identified, including FOXP4 and TYK2, which are linked to lung cancer and autoimmune diseases, respectively [33].

### 2.2. Agriculture

The ancient genetic modification that involved mainly selective breeding and artificial selection occurred more than 32,000 years ago, and the first artificially selected organism was thought to be the domesticated dog (*Canis lupus familiaris*), the descendant of grey wolf (*Canis lupus*) [109]. Although the early utilization of the genetically engineered technology ranged from drug discovery to the production of biorenewables, the most controversial application of the technology was and perhaps is for food production. The first crop to be genetically altered was the tomato (*Solanum lycopersicum*). The product Flavr Savr received marketing approval from the United States Department of Agriculture (USDA) in 1994 after years of extensive field experiments and health testing. The Flavr Savr was created by introducing a reverse-orientation copy of the polygalacturonase gene, which suppresses/shuts down the formation of the polygalacturonase enzyme that dissolves cell-wall pectin in conventional tomatoes, allowing the crop to stay firm longer after harvest [110].

Apart from extending the shelf life of food, the genetically engineered technology has been used to produce pesticide-resistant (or tolerant) plants that are easier to manage and cultivate. Two remarkable examples are Bt maize (*Zea mays*) and Bt cotton (*Gossypium hirsutum*) which have become the predominant varieties grown in the United States since their establishment in the mid-1990s. The genes of delta-endotoxins (δ-endotoxins) from the soil bacterium *Bacillus thuringiensis* encode Cry proteins which are specifically toxic to certain insect orders such as Lepidoptera, Diptera, and Coleoptera [111]. Nonetheless, Bt plants have been reported to be highly vulnerable to certain insect pests that proliferate in some countries such as India, making farming in these countries more capital-intensive [112]. Genetically engineered herbicide-resistant crops have also been created to control unwanted plants in fields efficiently, with the most eminent examples being the glyphosate (N-(phosphonomethyl)glycine)-resistant crops [113]. Glyphosate is considered as a broad-spectrum herbicide given that its presence would prevent almost all plants from making the essential proteins required for their survival.

Biofortification through rDNA or metabolic technology has also been attempted to increase the nutrition value of some staple crops [114]. Golden rice, for example, was developed to combat vitamin A deficiencies in developing countries [115]. In nature, the machinery to synthesize beta (β)-carotene (provitamin A) is fully active in rice leaves but partially turned off in its grains. The pathway is turned back on in golden rice with the addition of two genes encoding phytoene synthase (psy) and carotene desaturase (crtI) via genetic engineering, allowing β-carotene to accumulate in the grains [116]. In July 2021, the Philippines became the first country to approve the commercial production of the golden rice. It was recently reported that multiple biofortification traits (such as high provitamin A, high iron, and high zinc) can be introduced through metabolic engineering via transgenic technology. However, there have been no reported examples of sufficient nutrient enhancement through genome-editing approaches to date, and the combination of genetic engineering and conventional breeding is considered the most powerful approach when aiming at multi-nutrient crops [114].

The genetics of the first transgenic animal were successfully altered way back in 1986 by inserting a portion of the SV40 virus and herpes simplex virus (HSV) gene which encodes thymidine kinase (TK) into an early-stage mouse (*M. musculus*) embryo to develop cancer [117]. Since then, OncoMice have been a typical model organism in clinical studies. The latest tools to create transgenic animals for human disease studies, including CRISPR/Cas9 systems and transcription activator-like effector nuclease (TALEN), are summarised in Volobueva et al. [118]. Many species of animals have been genetically engineered to fend off pollution and starvation, from transgenic pigs (*Sus scrofa domesticus*) capable of producing an environmentally friendly form of manure to transgenic salmon (*Salmo salar*) that grow to a marketable size within 1.5 instead of 3 years [119]. Presently, there have been no genetically engineered animals approved to enter the human food chain, although the first biopharmaceutical product produced by genetically engineered goats (*Capra aegagrus hircus*) called the ATryn, an anticoagulant to treat a rare blood clotting disorder, was approved in 2009 [120,121].

The world may be on the brink of agreeing on the production of the first genetically engineered animal for human consumption, but the debate surrounding controversy in both animal transgenic and cloning technologies will live on. This is the case for the far-famed Dolly the Sheep (*Ovis aries*), the first mammal successfully cloned following somatic cell nuclear transfer from an established cell line [122]. Dolly was revealed to the world in 1996 and her death six years later was as controversial as her life when she only managed to live for about half of the life expectancy for sheep [123]. The announcement of the birth of the first gene-edited, cloned macaque monkeys (*Macaca fascicularis*) using CRISPR for Brain and Muscle ARNT-Like 1 (BMAL1) knockout has sparked outrage from animal welfare advocates and researchers around the globe [124,125]. Nonetheless, genetically engineered technologies can be beneficial if they are done right. A recent study reported several genetically engineered mouse models that may be useful for SARS-CoV-2 research to combat COVID-19 [126].

### 2.3. Biodiversity

The successful development of genetically modified bacteria (*E. coli*), more specifically GE insulin in 1973, marked the first breakthrough of the technology in the field of medicine [127], leading to the production of the diabetes drug Humulin. In the 1980s, several other bacteria were being genetically engineered. One notable example is the genetically modified *Pseudomonas putida*, which can help in oil spill mitigation with its ability to break down multiple components of crude oil [128]. During the last decade, genetically modified bacteria have been used to produce various bio-based products, including recombinantly produced chymosin (or rennin) in cheese production and the first-generation bioplastics and biofuels [129,130]. The genetic manipulation of the model cyanobacterium *Synechocyctis* sp. PCC6803, for example, has led to an increase in production of bioplastic polyhydroxybutyrate through overexpressions of Rre37 and SigE; the two major proteins involved in polyhydroxybutyrate synthesis [131]. More recently, the most abundant polyester plastic polyethylene terephthalate was successfully engineered to break down and recycle bottles [132].

Climate change, one of the defining issues of the 21st century, is predicted to be the cause of extinction for up to 40% of existing species in the next 30 years [133,134]. Biodiversity, as well as evolution and conservation, are becoming increasingly important as a result of climate change and habitat loss that can lead to extinction [135,136]. Facilitated adaptation, where gene variants from a well-adapted population are transferred into the genomes of threatened populations of either the same or different species, has been set forth to mitigate maladaptation and avert extinction. Nevertheless, this intervention may benefit only certain species and carries its own set of challenges and complications [137]. While some reports stated that climate-related local extinctions have occurred in hundreds of species, the equivalent number of species have merged to survive at their warm-edge range. This implies that genetic adaptations or phenotypic plasticity may enable some populations to tolerate warmer conditions [138]. Intraspecific adaptations should, therefore, be taken into account when assessing species’ vulnerability to climate change [139], though the prevailing issue is the absence of robust methodologies that fully allow the incorporation of genomic information in projecting species responses to a changing climate and in strategizing conservation plans [136,140]. Some species may survive climate change by either dispersing or niche shifts, or both [140,141].

International conservation policy recognises three levels of biodiversity: genetic, species, and ecosystem, all of which should be retained by conservation management [142]. Over the past three decades, many of the genetic, ecological, and geographical factors that contribute to species speciation have been well established, mainly due to the maturation of both theoretical and empirical speciation research [143]. One recent example is the study on the dynamics of explosive diversification and accumulation of species diversity based on the assembly of 100 cichlid genomes [144]. The rapid succession of speciation events within explosive adaptive radiation was reported to depend primarily on the exceptional genomic potential of the cichlids, which is driven by the high density of ancient indel polymorphisms that are mostly linked to ecological divergence [144]. Nonetheless, it is worth noting that the loss of genetic diversity in both terrestrial and marine ecosystems has accelerated during the last few decades, spurred largely by anthropogenic activities such as agriculture and industry [145]. Maintaining resilience, community function, evolutionary potential, and adaptive capacity in these ecosystems through the maintenance of genetic diversity is among the central components of the Sustainable Development Goals (SDGs), including SDG 14 Life Below Water and SDG 15 Life On Land. According to Jung et al. [136], spatial guidance is required to determine which land areas can potentially generate the greatest synergies between biodiversity conservation and nature’s contributions to humanity in order to support goal setting, strategies, and action plans for the biodiversity and climate conventions.

## 3. Genetic Revolution in the 21st Century: The Polemic of Gene Editing

Since its beginnings in the mid-2000s, the NGS has ignited the world of genomics with its refined approach to DNA sequencing. Unlike its predecessor capillary sequencing, which can sequence only a few DNA fragments, NGS is capable of generating millions of fragments simultaneously with automated library preparation [146]. Today, enormous data generated from NGS have been made available for multiple research areas, especially in personalised medicine, with the volume of data increasing on a day-to-day basis. These data can be used to uncover the causes of genetic diseases, although the process of tracking down specific disease-causing genes has not been straightforward, due to the complexity of the 3.2 gigabyte human genome [147]. This problem has been overcome by the invention of CRISPR, which exploits ancient bacterial immune systems to edit genes not only in humans but also in all other living things [19,148,149].

CRISPR was first discovered in 1987, when some unusual repeating sequences were found in the DNA of *E. coli* and several other bacteria, although the real benefit of CRISPR sequences remained mostly undiscovered until 20 years later when these odd sequences were found to play a vital role in the immune system of bacteria [150]. Since then, researchers have been patching the knowledge gaps, from detailing how spacer sequences derived from phage are transcribed into CRISPR RNAs (crRNAs) that guide Cas (CRISPR associated) proteins to the target DNA [151], to resolving that certain CRISPR systems can target DNA instead of RNA, or both [152,153] (Figure 2). The details on how the CRISPR/Cas immune system can adapt remarkably to cleave invading DNA were then reported, leading to the notion that the presence of the Cas enzyme has been the key to the survival of bacteria against viral infections for millennia [149,154]. The first Cas enzyme discovered was the Cas9 from *Streptococcus pyogenes*, which is also currently the most prominent and heavily utilized genome-editing tool. Other CRISPR family enzymes that are commonly used for genetic manipulation include Cas12 and Cas13 (Figure 2).

Recent years have witnessed the wide application of the CRISPR/Cas system in biomedical research, from reducing the severity of deafness to treating sickle cell anaemia and B-thalassemia in the model organism mouse, alongside reducing the severity of genetic deafness in them [155,156,157]. Some important examples of the use of CRISPR in dissecting or correcting inherited diseases such as Alzheimer’s and Parkinson’s can be found in Raikwar et al. [158]. Because the CRISPR/Cas is a versatile system that can cut up any genome in any location, it is considered a plausible tool in cancer therapy and drug discovery, such as new antibiotics and antivirals [149,159,160,161]. A CRISPR-Cas12-based assay was recently developed to detect SARS-CoV-2 from extracted patient sample RNA within 40 min, providing rapid yet accurate results with 95% and 100% positive and negative predictive agreements, respectively [25]. CRISPR-based diagnostics have indeed evolved briskly during the last half-decade, from merely a nucleic acid sensing tool to a clinically relevant diagnostic technology for the rapid, affordable, and ultrasensitive detection of biomarkers for routine clinical care [162].

Agriculture and biodiversity can also be benefited from the system; for example, in speeding up the development of climate-resilient crops, wiping out a disease-carrying mosquito species, or resurrecting extinct species (or gene drives that could alter entire species) [163,164,165]. Although there are endless possibilities for CRISPR to create a world without disease and hunger, the system is still riddled with flaws and far from perfect [166]. On a positive note, there are existing gene-editing tools developed during the pre-CRISPR era that can deal with some of the technical issues surrounding CRISPR. An outstanding example is the transcription activator-like effector nucleases (TALENs). TALENs is seen as a better tool than its predecessors, such as endonucleases and zinc-finger nucleases (ZFNs), due mainly to its high degree of precision and control [167]. The differences and applications of these tools are presented in Table 2. It is recognized as a more appropriate tool for healthcare compared to CRISPR and is preferred by top cancer immunotherapy-focused companies (such as Cellectis and Adaptimmune) due to its higher efficiency in scaling up edited genomes, and more importantly, its lower risk of off-target cleavage events.

While the most significant backlash against gene-editing technology has been with regard to the production of genetically modified foods, the greatest controversy surrounding CRISPR is perhaps its utilisation in human germline gene editing [175,176]. The first successful case of the creation of human babies with CRISPR-edited genomes—two girls resistant to HIV—has stunned the scientific world and raised many safety and ethical concerns [176,177]. The fundamental question is, what are the dos and don’ts of utilizing CRISPR? Whilst there are some important distinctions being drawn in this debate, perhaps the most important is the distinction between somatic and germline gene editing [178,179,180,181]. Most changes introduced by gene editing are limited to somatic cells, which are all cells in the body except for the germline (egg and sperm) cells. Changes made through somatic germline editing only affect certain tissues and the altered DNAs are not passed to the next generation [182]. When the change ends with the person being treated, it is considered purely medicinal. In Figure 3, the top left triangle depicts treatments for people living with diseases, and the main challenge here is delivery.

Different diseases affect different cells in the body and some organs have easier deliveries, and vice versa [183,184]. Genetic disorders or conditions that affect the blood or the bone marrow (such as sickle cell disease and human immunodeficiency virus) might deliver most promising CRISPR experiments, given that those abnormal cells can be removed from the patient’s body, edited externally, and replaced in the patient only after the editing is confirmed. Nevertheless, one major challenge of somatic gene editing is to strike a balance between therapy and enhancement (Figure 3). In general, therapies treat diseases while enhancements involve phenotypic improvement of healthy people. Studies have found rare genes linked to a lower risk of some conditions such as diabetes and heart disease: should the introduction of these genes to a healthy person be considered therapy or enhancement?

Somatic gene editing has always been much less controversial compared with germline gene editing, which often passes the altered DNA of the germline cells or embryos to future generations, making modifications that ultimately affect the human population and evolution [179,180,181,182]. Both germline cell and embryo genome editing remain illegal in many countries because they trigger many ethical and safety concerns [178,185,186]. Like somatic gene editing, the line between therapy and enhancement in germline gene editing is unclear, especially when it comes to genetic conditions that are not detrimental or harmful to the carriers—for instance, achondroplasia that results in dwarfism [187]. Would it be ethical to edit the defective fibroblast growth factor receptor-3 (FGRGR3) gene in human embryos to eventually eliminate the non-lethal form of short-limbed dwarfism in humans? The possibility to have gene-edited (or designer) babies, where parents get to pick traits that they like or deem superior (such as intelligence, athletic ability, height, and eye colour) for their children, is an extreme example of the therapy (Figure 3) [188,189]. Some of these traits are complex traits that involve high-order genetic interactions, partly nature (genes) and partly nurture (environment), and this could potentially make them bad targets for immature tools like CRISPR [190,191,192]. Nonetheless, these tools may be useful for understanding and controlling biological function with effective optimisation [193].

In fact, the nature versus nurture debate has long been argued since the early ages until the interest in epigenetics was revived during the early 1990s. The term epigenetics, first coined by Conrad Waddington in 1942 [194], describes the heritable changes in organisms through gene expression rather than the alteration of the genetic code. In other words, epigenetics can explain the heritable changes in a phenotype that occur without changes to the genotype. DNA methylation is currently one of the most widely studied epigenetic modifications [195], along with a handful of other major modifications including histone modifications, chromatin remodelling, and microRNA expression [196]. The epigenetics field has grown rapidly over past decades, revealing that environmental and individual lifestyle factors can influence epigenetic mechanisms. For example, prenatal factors such as mode of conception and maternal diet have been found to affect the DNA methylation of several type 2 diabetes-related genes, notably the insulin-like growth factor II (IGF2) in children that are prenatally exposed to famine conditions [197,198]. Additionally, individuals with prenatal famine exposure have been reported to be more prone to schizophrenia [199,200].

The increased knowledge of epigenetics, coupled with advanced technologies such as NGS and CRISPR, will aid in the development of improved molecular diagnostics and targeted therapies across the clinical spectrum [193]. For example, an enhanced epigenetic profiling conducted by Shchukina et al. [201] revealed a specific signature of heathy aging in human classical monocytes, indicating that the gradual age-associated changes of DNA methylation could be the reason certain diseases (such as adult-onset asthma) develop later in life. Epigenetic ageing in long-lived animal species, particularly bats, demonstrated that enhanced epigenetic stability may facilitate their exceptional longevity, particularly in cases where innate immunity and cancer suppression genes are involved [202].

## 4. Future Perspectives

COVID-19 paints a bleak picture of our world’s vulnerability to unforeseen complex global risks and crises, emphasising the significance of being proactive and developing new research to sustain it. Reminiscing on previous decades of heredity research, genetics has come a long way and has established itself as a household name that will shape the future of humanity and sustainability. As the general public has a better grasp of the relevance of genetics in their own lives, so do debates over the flaws and foibles that come with its technologies. The news headlines from 2018 about the creation of the world’s first CRISPR-edited babies with artificially increased resilience to HIV have shaken the scientific world indefinitely [203]. The experiment, which edited the gene C-C chemokine receptor type 5 (CCR5), was not performed properly, leading to mosaicism, and the accusation of being profoundly disturbing and trampling on ethical norms [177,204]. Additionally, recent studies have indicated that CRISPR-edited cells can inadvertently trigger cancer, alongside rearranging or eradicating important DNA that could imperil human health [161,205].

There is no doubt that human gene-editing experiments are generally premature, and the risks and uncertainties associated with them are high [203,206,207]. As technology advances at a breakneck pace, there are more unresolved issues and challenges surrounding gene therapy and the clinical products accessible today. Only a year after Jennifer Doudna and Emmanuelle Charpentier, the two pioneers of the revolutionary CRISPR tool, were awarded the 2020 Nobel Prize in Chemistry [208], a newer, superior alternative editing technology, Retron Library Recombineering (RLR), was introduced [209]. RLR, which can generate millions of mutations simultaneously, is regarded as a simpler, more flexible gene-editing tool that eliminates the toxicity frequently associated with CRISPR and improves researchers’ ability to investigate mutations at the genome level [209]. Nevertheless, it is worth noting that before gene-editing tools such as CRISPR and RLR, other developed genetic technologies (such as in vitro fertilization, embryo cryopreservation, and preimplantation genetic diagnosis) have undergone similar criticisms, before being publicly approved and implemented on a large scale. The long-term consequences of these technologies were also unclear when they were first introduced. The similarity between all these cases is that the researchers involved had the ultimate goal of improving the lives of the parent-to-be and the future child by allowing the selection of certain attributes and eliminating any harmful limitations imposed by the natural bloodline.

Actively solving any issues surrounding new tools instead of delaying or stopping clinical studies would be the best way forward [207]. Lessons could be learned from the Gelsinger case. This instigated one of the major setbacks in gene therapy, when the 18-year-old Jesse Gelsinger did not survive the clinical trials aimed at treating his ornithine transcarboxylase deficiency (OTCD)—an X-linked recessive disorder that prevents the body from regulating ammonia levels [210,211]. As a result, progress in gene therapy development was then hampered for several years due to public reaction, which was unnecessary because individual adverse reactions can happen for any treatments. Clinical trials on different individuals are unpredictable, and no medical specialist can claim that a treatment is absolutely safe. However, if we were to use tools such as CRISPR for medical enhancement instead of therapy, it could be questioned whether it would be better and simpler to inject hormones to promote growth in children, rather than editing their genes, which cannot be undone, and carries many unknown consequences. Before CRISPR, it was often difficult and costly to develop new antibiotics for deadly infections [161]. Theoretically, the CRISPR/Cas9 systems could be used to eradicate certain bacteria precisely, although establishing the delivery mechanisms could remain a challenge. Excessive precaution will only prevent significant benefits from newly developed tools for the foreseeable future.

In the field of agriculture, new tools like CRISPR should be made transparent and public trust should be built before genome editing is conducted, to avoid the same mistakes being made with GMOs [212,213]. This needs to be achieved soon, as scientists are already looking to edit important genes of various crops and farm animals, from developing bananas that are resistant to deadly fungal disease, to creating hornless cattle [163,214,215]. In fact, major companies, such as DuPont and Monsanto, have already begun licensing CRISPR technology to develop new and improved crop varieties. Currently, foodstuffs that have genes knocked out via CRISPR seem to be regulated more lightly than GMOs. While CRISPR may not replace techniques like GMO entirely, it can help identify targeted genes much more quickly, as well as have the ability to insert desired traits into crops more precisely than traditional breeding. In fact, the CRISPR-Cas has been widely used in agriculture and plant biotechnology in the past decade, from increasing plant yield and quality to breeding and accelerated domestication [216].

Down the road, it might be difficult to keep rogue experiments in check, considering the ease of use and low costs of newly developed tools such as CRISPR. If these tools are being utilised ethically and innovatively, it is not far-fetched to consider them safe to be used on humans, although the main concerns will always be whether we have enough knowledge to prevent harm and only do definite good. Some technologies have avoided the ethical issues that plagued their predecessors; for example, embryonic stem cells have been superseded by iPSC, which treats human diseases by reprogramming cells from adult tissues into an embryonic-like state. CRISPR/Cas9 genetic modifications, biomaterials, and 3D printing have all enhanced the capabilities of iPSCs [217]. Large-scale, superfast, and low-cost DNA sequencing has allowed human genomes to be mapped within hours for merely USD 1000, driving genomics into mainstream medicine, placing the patient first, rather than the disease, at the centre of healthcare. Genomics is enabling more precise and personalised disease prediction, prevention, diagnosis, and treatment based on a person’s unique biochemical makeup, ushering in a revolutionary shift toward precision medicine. It won’t be long before every cancer patient’s genome is sequenced, and drugs that specifically target their genetic alteration can be developed. Genomics also holds the promise of modifying genes in embryos and eradicating inherited diseases.

However, because biology is variable and complex, a 100% prediction may be challenging to attain, even if all genes, protein structures, and regulatory sequences of an organism are known. It is worth noting that protein coding genes make up only about 1.5% of the human genome, with the remainder thought to be junk with no discernible function. Furthermore, as the Earth’s temperature continues to shift more unpredictably as a result of global warming, the environmental impact becomes increasingly significant for more complex free-ranging organisms. Because many external factors influence how a genome works, general features of the epigenetic landscape and transcriptional output will need to be routinely incorporated into predictive models for calculating the effect of genotype on phenotype. The current main goal of the World Health Organization is to help every human on earth in achieving the highest possible standard of health, while the SGDs were developed to achieve a better and more sustainable future for all. After all, traditional tools of evidence-based preventive medicine or breeding are unlikely to be able to sustain livelihoods and food security amidst rapidly growing populations and climate change. The use of advanced tools such as gene editing and GMOs could be a viable way for humanity to progress.

## Figures and Tables

**Figure 1 ijms-23-03976-f001:**
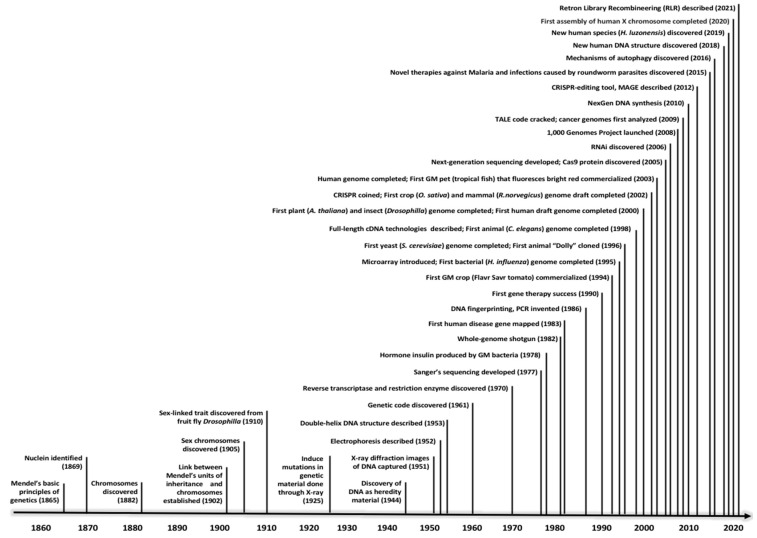
Notable genetic discoveries in the past one and a half centuries.

**Figure 2 ijms-23-03976-f002:**
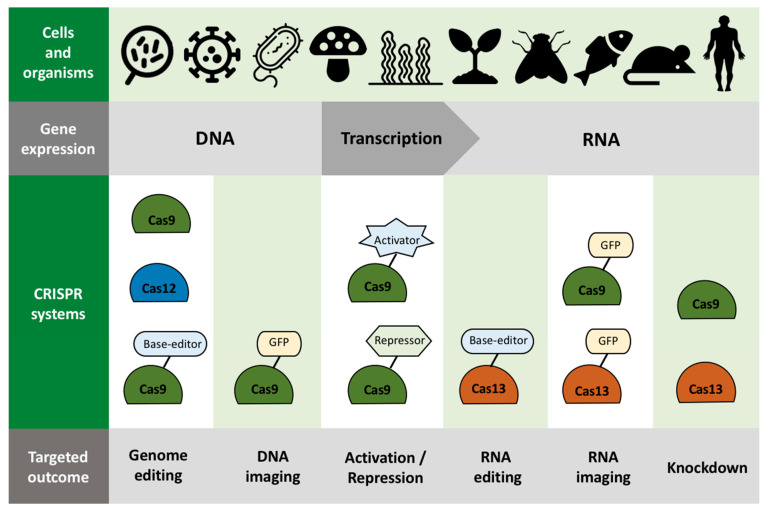
Common CRISPR/Cas systems used for genetic manipulation.

**Figure 3 ijms-23-03976-f003:**
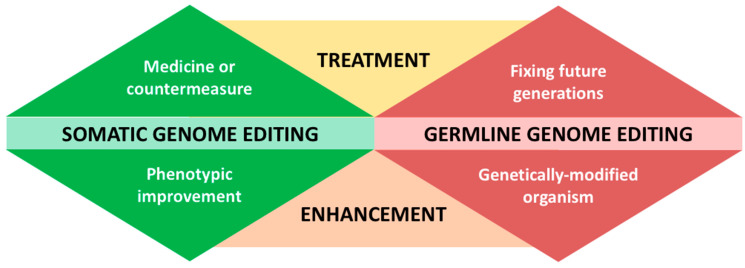
The differences between somatic and germline genome editing.

**Table 1 ijms-23-03976-t001:** Examples of multicellular organisms with well-annotated genomes.

**Kingdom**	**Species**	**Relevance**	**Estimated Genome Size (Mbp)**	**Reference**
Animalia	Aedes mosquito (*Aedes aegypti)*	Primary vector for yellow and dengue fevers	1380	[37]
	Cattle (*Bos taurus*)	Ruminant biology and evolution	2870	[38]
	Coelacanth (*Latimeria chalumnae*)	Tetrapod evolution	2860	[39]
	Common chimpanzee (*Pan troglodytes*)	Model organism (human population genetics and evolution)	2400	[40]
	Common marmoset (*Callithrix jacchus*)	Biomedical research application	2260	[41]
	Giant panda (*Ailuropoda melanoleuca*)	Foundation for promoting mammalian genetic research	2250	[42]
	Honeybee (*Apis mellifera*)	Model organism (social behaviour and global ecology)	1800	[43]
	Japanese medaka (*Oryzias latipes*)	Vertebrate evolution	700	[44]
	Pacific oyster (*Crassostrea gigas*)	Lophotrochozoa evolution	559	[45]
	Platypus (*Ornithorhynchus anatinus*)	Model organism (combination of reptilian and mammalian characters)	1840	[46]
	Red flour beetle (*Tribolium castaneum*)	Model organism (beetle and pest)	160	[47]
	Sea urchin (*Strongylocentrotus purpuratus)*	Model organism (developmental and system biology)	814	[48]
	Sponges (*Amphimedon queenslandica*)	Animal origins and early evolution	167	[49]
	Two-spotted spider mite (*Tetranychus urticae*)	Cosmopolitan agricultural pest	90	[50]
	Western gorilla (*Gorilla gorilla*)	Human origins and evolution	5400	[51]
	Mexican oxolotl (*Ambystoma mexicanum*)	Evolutionary changes in key tissue formation regulators	32,000	[52]
	Galapagos cormorant (*Phalacrocorax harrisi*)	Evolutionary changes in the size and proportion of limbs	1200	[53]
	Golden orb-weaver (*Nephila clavipes*)	Diversity of spider silk genes and their complex expression	2440	[54]
Plantae	African oil palm (*Elaeis guineensis*)	Oil-bearing crop	1800	[55]
	Amborella (*Amborella trichopoda*)	Angiosperm evolution	870	[56]
	Barrel medic (*Medicago truncatula*)	Model organism (legume)	246	[57]
	China rose (*Rosa chinensis*)	Model organism (ornamental plant)	560	[58]
	Dwarf banana (*Musa acuminata*)	A genome of modern cultivar	523	[59]
	Maize (*Zea mays*)	Major cereal crop	2300	[60]
	Papaya (*Carica papaya*)	Tropical fruit crop	372	[61]
	Peanut (*A.* *duranensis*, *A. ipaensis*, *A. hypogaea*)	Polyploid genetic mechanisms	2540	[62,63]
	Pigeon pea (*Cajanus cajan*)	Model organism (legume)	833	[64]
	Potato (*Solanum tuberosum*)	Major root crop	844	[65]
	Quinoa (*Chenopodium quinoa*)	Future crop	1500	[66]
	Rose gum (*Eucalyptus grandis*)	Fibre and timber crop	640	[67]
	Sorghum (*Sorghum bicolor*)	Major cereal crop	730	[68]
	Soybean (*Glycine max*)	Major protein and oil crop	1115	[69]
	Tomato (*Solanum lycopersicum*)	Major vegetable crop	900	[70]
	Silver birch (*Betula pendula*)	Model organism (forest biotechnology)	440	[71]
	Durian (*Durio zibethinus*)	Tropical fruit biology and agronomy	738	[72]
	Sunflower (*Helianthus annuus*)	Oil metabolism, flowering, and Asterid evolution	3600	[73]
	Tausch’s goatgrass (*Aegilops tauschii*)	Genetic resources for wheat	4300	[74]
	Barley (*Hordeum vulgare*)	Major cereal crop	4800	[75]
	Pearl millet (*Pennisetum glaucum*)	Future crop	1790	[76]
Fungi	Black mold (*Aspergillus niger*)	Model fungal	34	[77]
	Filamentous fungus (*Aspergillus nidulans*, *A. fumigatus*, *A. oryzae*)	Model fungal	40	[78]
	Fission yeast (*Schizosaccharomyces pombe*)	Model yeast	14	[79]
	Rice blast fungus (*Magnaporthe grisea*)	Model fungal	40	[80]
	Split gill (*Schizophyllum commune*)	Model mushroom	39	[81]
	Yeast (*Candida albicans*)	Human pathogen	4	[82]
	Filamentous fungus (*Penicillium chrysogenum*)	Industrial use	32	[83]

Sources: [84,85].

**Table 2 ijms-23-03976-t002:** Comparison of core genome-editing tools in the 21st century.

Property	CRISPR/Cas9	TALEN	ZFN	Meganuclease (Homing Endonucleases)
Essential components	sgRNA and Cas9	TALE and FokI	ZFP and FokI	Meganuclease (nuclease domain)
Backbone origin	Bacteria (*Streptococcus pyogenes*)	Bacteria (*Xanthomonas* spp.)	Mostly prevalent in eukaryotes	Microbial mobile genetic elements
Ease of engineering	Easy; facile design of gRNA, standard cloning methods and oligo synthesis	Moderate; complex cloning methods are required	Difficult; substantial protein engineering is required	Difficult; substantial protein engineering is required
Recognition site	22 bp (20-bp guide sequence and 2-bp protospacer adjacent motif (PAM) for Cas9); 44 bp for double nicking	28–40 bp per TALEN pair	18–36 bp per ZFN pair; guanine-rich region	14–40 bp
Specificity	Highly predictable (DNA–RNA interaction). Multiple mismatches tolerated	Less predictable (DNA–protein interaction). Small number of mismatches tolerated	Less predictable (DNA–protein interaction). Small number of mismatches tolerated	Less predictable (DNA–protein interaction). Small number of mismatches tolerated
Targeting constraints	Targeted sequence must precede PAM	T must be the 5′ targeted base for each TALEN monomer	Non-G-rich sequences are difficult to target	Low efficiency in targeting novel sequences
Ease of in vivo delivery	Moderate	Difficult	Relatively easy	Relatively easy
Multiplexing ability	Feasible	Challenging	Challenging	Challenging
Affordability	Highly affordable (1–3 days)	Affordable but time consuming (5–7 days)	Resource intensive and time consuming (7–15 days)	Resource intensive and time consuming (up to 100 days)
Methylation sensitivity	No	Sensitive	Sensitive	Sensitive
Clinical or pre-clinical stage	Clinical trial application for refractory non-small-cell lung cancer, sickle cell disease, and beta-thalassemia	Clinical trial application for relapse or refractory acute myeloid leukaemia	Clinical trial application for HIV and Hunter’s syndrome	Clinical trial application for non-Hodgkin Lymphoma and multiple myeloma

Sources: [168,169,170,171,172,173,174].

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
