# Peer review of "Genetics Matters: Voyaging from the Past into the Future of Humanity and Sustainability"

_ijms, 2022, doi:10.3390/ijms23073976_

Round 1
Reviewer 1 Report
In this article, Cheng et al. give a comprehensive review of the history of genetics. It is a well-written article that coherently reviews each of the points made by the author. For me it is an article that should be accepted, however I would like to highlight a few minor points that the authors should review before acceptance.
- In line 39 there is a sentence about the first assembly of the X chromosome. I think that the phrase is not in the right context. In this paragraph the authors are referring to 19th and 20th century discoveries. Possibly, at some other point in the article it would make more sense.
- In the introduction, the paragraph from line 81 to 99 seems to me unnecessary. It is a discussion about the prices and names of the technologies used for NGS that, if needed, could be summarized in 3-4 lines.
- The article reviews several fields where genetics has played a fundamental role during the last decades, especially in the field of agriculture and medicine. Although, I miss some reference to other fields such as conservation or evolution, where genetics has also played (and still plays) a key role.
- In line 342, the authors define the CRISPR technique after having mentioned it previously in line 103. Possibly, the first mention to CRISPR and SARS-Cov-19 findings could be eliminated and merged with paragraph 361-373, where the authors already discuss this technique in more detail.
- Line 377 "another example is it could one day be used to wipe out the entire population of malaria-spreeding mosquitoes". I miss here some reference or some more detailed explanation about the example they propose. The way this sentence is written, I find it difficult to understand its meaning.
- line 407. "genetic disorders or conditions that live in the blood or the bone marrow". Is this correct? The word "live" in this context leads to confusion.
- Finally, and as a relevant point. It seems to me that the discussion is very focused on praising the advantages and benefits of gene editing, leaving aside its unresolved weaknesses and ethical criticisms. The authors even comment in line 380 that it is a technique that is far from perfect. With which I agree. Thus, as an example, the last sentence "is one of the few possible, feasible ways for humankind to move forward" is very exaggerated and should be lowered.
Author Response
We greatly appreciate the comments from the Reviewers which we have addressed accordingly in the attached document.

Reviewer 2 Report
The authors presented a good review piece with a clear focus on the development of gene-editing tools over time.
I would like to recommend that the review focus only on the medical angle only. It was confusing and out of context to include the agricultural angle. It was not well incorporated and out of place.
Some minor comments:
line 68: This sentence is not really falling well with the previous info. Maybe move it to the next para?
line 73: Add the year and more details about the Maxam-Gilbert sequencing.
line 74: Before jumping to sequencing, I think you need to talk about PCR as a technique?
Line 81: Maybe define the sequencing technology difference between Sanger and Next-generation sequencing.
Line 91: PCR should be all in caps. PCR is an abbreviation that you haven't defined or talked about earlier.
line 100: Since when exactly?
line 158: add a reference.
line 218: I would suggest deleting all that section.
line 287: I would suggest rewriting that section to be more concise and to get to the point clearer.
Author Response
We greatly appreciate the comments from the Reviewers which we have addressed accordingly. Our point-by-point responses are in the attached document.
